# Ultrasound stimulation of the motor cortex during tonic muscle contraction

Ian S. Heimbuch[1,2,3]*, Tiffany K. Fan[4], Allan D. Wu[5], Guido C. Faas[1], Andrew C. Charles[1], Marco Iacoboni[2,3]

1 Department of Neurology, David Geffen School of Medicine, University of California, Los Angeles, Los Angeles, California, United States of America, 2 Department of Psychiatry and Biobehavioral Sciences, David Geffen School of Medicine, University of California, Los Angeles, Los Angeles, California, United States of America, 3 Ahmanson-Lovelace Brain Mapping Center, University of California, Los Angeles, Los Angeles, California, United States of America, 4 David Geffen School of Medicine, University of California, Los Angeles, Los Angeles, California, United States of America, 5 Department of Neurology, Feinberg School of Medicine, Northwestern University, Evanston, Illinois, United States of America

* heimbuch@ucla.edu

**Data Availability Statement:** All EMG files and associated meta data have been uploaded to the Open Science Framework database (doi:10.17605/OSF.IO/W547Y). Code to run acoustic simulations

## Abstract

Transcranial ultrasound stimulation (tUS) shows potential as a noninvasive brain stimulation (NIBS) technique, offering increased spatial precision compared to other NIBS techniques. However, its reported effects on primary motor cortex (M1) are limited. We aimed to better understand tUS effects in human M1 by performing tUS of the hand area of M1 (M1$_{hand}$) during tonic muscle contraction of the index finger. Stimulation during muscle contraction was chosen because of the transcranial magnetic stimulation-induced phenomenon known as cortical silent period (cSP), in which transcranial magnetic stimulation (TMS) of M1$_{hand}$ involuntarily suppresses voluntary motor activity. Since cSP is widely considered an inhibitory phenomenon, it presents an ideal parallel for tUS, which has often been proposed to preferentially influence inhibitory interneurons. Recording electromyography (EMG) of the first dorsal interosseous (FDI) muscle, we investigated effects on muscle activity both during and after tUS. We found no change in FDI EMG activity concurrent with tUS stimulation. Using single-pulse TMS, we found no difference in M1 excitability before versus after sparsely repetitive tUS exposure. Using acoustic simulations in models made from structural MRI of the participants that matched the experimental setups, we estimated in-brain pressures and generated an estimate of cumulative tUS exposure experienced by M1$_{hand}$ for each subject. We were unable to find any correlation between cumulative M1$_{hand}$ exposure and M1 excitability change. We also present data that suggest a TMS-induced MEP always preceded a near-threshold cSP.

## Introduction

Transcranial ultrasound stimulation (tUS) has gained attention in the past years as a potential new tool for noninvasive brain stimulation (NIBS). tUS has higher spatial precision compared to other NIBS techniques such as transcranial magnetic stimulation (TMS) and transcranial

is available via the MATLAB package TUSX (www. tusx.org).

**Funding:** This work was funded by The Arline and Seymour Kreshek Family Fund. The funders had no role in study design, data collection and analysis, decision to publish, or preparation of the manuscript.

**Competing interests:** The authors have declared that no competing interests exist.

electric stimulation (TES), which presents a possibility of improved targeting [1–5]. Furthermore, tUS can deliver its energy much deeper while maintaining focal precision—deeper than TMS or TES [6].

Previous studies have demonstrated that ultrasound is capable of stimulating central structures in animals [7], peripheral nerve pathways in animals and humans [8, 9], the retina [10], and intact brain circuits in animals [11]. However, the number of human tUS studies thus far is limited. In primary somatosensory cortex, tUS has been shown to modulate touch discrimination [2], induce localized somatosensations when targeting the cortical hand representation [3, 12], and induce changes in intrinsic and evoked EEG dynamics [13]. In primary visual cortex, tUS can induce individual visual phosphenes, percepts of a flash of light, that were accompanied with an evoked potential and blood-oxygenation-level-dependent (BOLD) contrast similar to those seen with photic stimulation [14]. Likewise, the effects of tUS in primary motor cortex (M1) in humans are a focus of active investigation.

Much of our understanding of motor cortex stimulation comes from TMS investigations, where it has been the standard for noninvasive M1 perturbation for decades [15–20]. Suprathreshold single-pulse TMS of M1 induces contraction in the corresponding muscles, and electromyography (EMG) allows for the quantification of these motor evoked potentials (MEPs). Since MEP size increases as a sigmoidal function of TMS intensity above motor threshold [17, 21], MEP strength is frequently used as an indicator of corticospinal excitability, both in neuromodulatory and behavioral interventions [15–20, 22–25]. This is supported by pharmacological evidence that shows motor threshold, the TMS intensity needed to elicit an MEP, is a proxy for the within-subject excitability of the cortico-cortical axons affected by the induced current of TMS pulse [26]. We investigated the neuromodulatory effects of tUS on M1 by analyzing its effect on TMS-evoked MEP.

Cortical silent periods (cSP) are a phenomenon of suppressed EMG activity during tonic contraction following single-pulse TMS of the corresponding M1 motor representation. cSPs typically last from 100–300 ms. Importantly, cSPs are considered to be driven predominantly by cortical inhibition from ~50 ms after instigation [27]. Specifically, pharmacological evidence suggests that the cSP effect is mediated by GABA receptor-dependent postsynaptic inhibition [26, 28]. When elicited during tonic contraction of the contralateral hand, cSPs are reported to be observable either following an MEP or without inducing an MEP, at subthreshold TMS intensities [29, 30]. As such, cSP provides a valuable method of investigation of inhibitory mechanism in motor cortex.

As a whole, the field is still building its understanding of what, if anything, tUS can affect via M1 stimulation. To date, no tUS study has been able to induce motor contraction through human M1 stimulation. Given that change in motor contraction strength has been the benchmark for M1 modulation studies, the established capabilities of TMS have been leveraged to investigate tUS effects on M1. For example, tUS of the hand area reduces the size of motor evoked potentials (MEPs) evoked by concurrent and concentric TMS [31–33]. A separate study reported a lasting increase in the size of MEPs after exposure to an ultrasound imaging device [34]. Additionally, tUS of M1 alone was shown to affect reaction times in two motor tasks [31, 32].

Because of its physiological underpinnings, the cSP is in a unique position to be leveraged as an externally detectable phenomenon to better understand tUS effects on M1. Specifically, tUS has been proposed to preferentially affect inhibitory interneurons [2, 32, 35–38], feeding well into cSP's existence as a interneuron-facilitated phenomenon. Additionally, since cSPs have been reported to occur without a preceding MEP [27, 29, 30, 39, 40], tUS' apparent inability to instigate an MEP does not preclude its use to attempt induction of cSPs. But as of yet, it is unknown if tUS can engage the necessary inhibitory circuits to instigate a cSP. We addressed

this by performing tUS of M1 on participants executing voluntary muscle contraction and analyzing the EMG data from the contracted muscle.

## Methods

Two experiments were performed in this study. In Experiment 1, we measured how tonically contracted hand muscles respond to single-pulse TMS and single-burst tUS of M1. We performed separate trials using tUS and TMS. In Experiment 2, we measured if cortical excitability changes following tUS exposure. Excitability was gauged using single-pulse TMS before and after tUS exposure.

### Data acquisition

**Participant demographics.** Research participants were right-handed with no neurological conditions. Due to TMS use, subjects with an increased risk of seizure were excluded (S1 Text). Due to MRI use, subjects with MR-incompatible implants were excluded. Participants were 18 to 42 years old, A subset of Experiment 1 participants (n = 10; mean: 25.9 years) participated in Experiment 2 (n = 8; mean: 26.75 years). Note that subject ID numbers are not sequential since other recruited subjects were used in a different study. Participants provided written consent before participating, and the study protocol was approved by the Institutional Review Board of the University of California, Los Angeles (IRB#17–000958).

**EMG and NIBS placement.** Electrode sites were cleaned with abrasive skin prep gel (Nuprep) and alcohol wipes. A surface EMG electrode (two 10 x 1 mm contacts; 10 mm spacing) measured the right first dorsal interosseous (FDI) muscle activity, and the signal was amplified (x1000) (Delsys Inc., Boston, MA) and sampled at 5000 Hz. The surface electrode was additionally secured to the finger with medical tape. A wide ground electrode was placed on the back of the hand. EMG was recorded for 1-second epochs around NIBS (both TMS and tUS) onset. A structural MRI (T1-weighted; 0.8 x 0.8 x 0.8 mm voxels) was acquired in a previous visit, and each structural MRI was registered to standard space for NIBS targeting of standard-space coordinates (Montreal Neurological Institute, MNI; 1-mm atlas). Registration was performed in FSL (the FMRIB Software Library) on brain volumes extracted using the opti-BET tool for FSL's BET [41, 42]. NIBS position with respect to the subject's head was tracked using neuronavigation software (Brainsight, Rogue Research, Montreal, QC) loaded with the subject's MRI. The neuronavigation software was prepared with pre-determined trajectories, which were the shortest Euclidean distance from the scalp to a set voxel as determined by a custom MATLAB script (Mathworks, Inc., Natick, MA). At the beginning of each NIBS session, five single TMS pulses were given at each target of a 6-target grid over the left motor cortex using MNI space (S12 Fig). The grid's origin was placed at MNI coordinates that correspond to $M1_{hand}$ as based on a meta-analysis of fMRI motor experiments: x = −39, y = −24, z = 57 [43]. This coordinate corresponds morphologically to the cortical 'hand knob' [44]. The other five targets on the grid were in a 12 voxel-width grid (9.6 mm grid interval) anterior, posterior, and medial, anteromedial, and posteromedial from the $M1_{hand}$ coordinate in subject space. The targets that elicited the largest, second-largest, and third-largest average MEPs were used as placement points for the NIBS devices. These three positions are referred to below as "TMS target", "2nd-best", and "3rd-best" targets respectively. For all TMS trials, the TMS coil was oriented with the handle pointed backwards and angled 45˚ from midline.

**Experiment 1, cSPs.** Participants moved their index finger laterally during trials to maintain consistent FDI contraction across trials, as monitored by a digital scale (20–40% maximum voluntary contraction, depending on the subject). NIBS was delivered during contraction. Percent maximum contraction varied between subjects so that every subject

maintained a comparable level of EMG activity (~150–200 peaks per second). 20 trials were performed at each of three TMS intensity levels: 90%, 100%, and 110% of % active motor threshold (aMT) (60 TMS trials total per subject). 20 tUS trials were performed for each of the following four parameters: 300-ms burst duration at the TMS target, 300-ms burst duration at the 2nd-best target, 300-ms burst duration at the 3rd-best target, and 500-ms burst duration at the TMS target (80 tUS trials total per subject) (S18 Fig). Subjects were told to use the feedback of the digital scale display to maintain their target FDI contraction force, and subjects were cued to relax between trials to avoid fatigue. Subjects monitoring their contraction level also had the benefit of keeping subject attention constant across trials in Experiment 1, since attention can affect EMG measurements [45, 46]. Trials had a jittered intertrial interval of 10 ± 2 s. The order of tUS vs. TMS-cSP blocks were not counterbalanced since counterbalancing Experiment 1 blocks would have conflicted with acquisition of Experiment 2.

**Experiment 2, Cortical excitability.** MEPs were measured with the subject's hand relaxed. TMS was delivered at the same suprathreshold intensity for both "before" and "after" conditions within subjects. TMS was set to 110–120%rMT (percent resting motor threshold), (110%: n = 1; 115%: n = 4; 120% n = 3). %rMT was varied across subjects to assure that each subject had consistent MEP sizes. 20 MEPs were acquired both before and after the tUS exposure protocol. tUS exposure protocol was the same as described in "Experiment 1, cSPs:" (80 total trials: 20 trials each: 300 ms at TMS target, 300 ms at 2nd-best, 300 ms at 3rd-best, 500 ms at TMS target) (S18 Fig). Trials had a jittered intertrial interval of 10 ± 2 s.

Participants of Experiment 2 (n = 8) also participated in Experiment 1 (n = 10). For participants of Experiment 2, Experiments 1 and 2 were performed during the same visit (S19 Fig). Performing Experiments 1 and 2 at the same visit with the same participant allowed us to minimize the number of volunteers who would be exposed to tUS.

**tUS equipment.** The tUS device used was a 500-kHz focused piezoelectric transducer (Blatek Industries, Inc., State College, PA). The transducer was manufactured with a face width of 3 cm and a focal point of 3 cm. When measured in water, the transducer produced a focus (-3 dB focus area) centered at 29.6 mm that was 2.8 mm wide and 20 mm long (-6 dB focus: centered at 33.7 mm, width 6 mm, length 33 mm). The transducer was housed in a custom 3D-printed handle, and an infrared tracker was mounted to the housing for neuronavigation (Fig 1). The transducer was driven by 500-kHz sine-wave voltage pulses from a waveform generator (33500B Series, Keysight Technologies, Santa Rosa, CA) and voltage pulses were amplified by a 50-dB radio frequency amplifier (Model 5048, Ophir RF, Los Angeles, CA). A 3-dB fixed attenuator was attached in line following the amplifier. Subjects' hair was parted at the location of the tUS targets to reduce the hair's effect on acoustic propagation, and ample ultrasound gel was used to acoustically couple the transducer to the subject's head. The cooling fans of the server rack-mounted radio frequency amplifier were quite loud, which acted as an auditory mask throughout the duration of tUS session. Cooling fan noise was constant throughout the duration of all tUS sessions.

tUS bursts were pulsed with a 1-kHz pulse repetition frequency and a duty cycle of 36% (Fig 1). Each burst lasted either 300 or 500 ms (tUS on for 108 or 180 ms total). Transducer output was confirmed via measurements made via hydrophone in degassed water (1 mm, Precision Acoustics Ltd, Dorchester, UK). Transducer output was set to produce an intensity of 15.48 W/cm$^2$ in degassed water (spatial peak, pulse average; $I_{sppa}$); spatial peak, temporal average ($I_{spta}$): 5.57 W/cm$^2$. Using previously measured attenuation of a 500-kHz ultrasound through human skull (-12 dB to -16 dB) [47], these parameters produce an estimated intracranial $I_{sppa}$ of 3.9 W/cm$^2$ ($I_{spta}$ = 1.4 W/cm$^2$). This is within safe levels (max Mechanical Index: 0.8) [48] and is within intensities of previous human tUS studies [2, 3, 13, 14, 49].

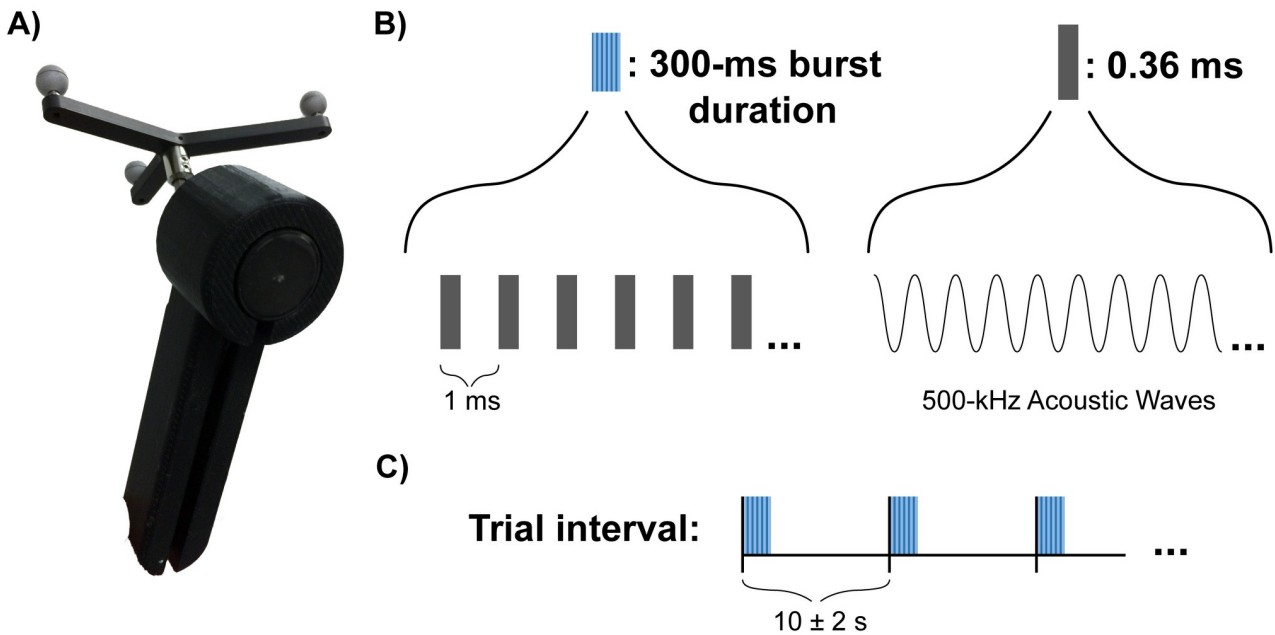

**Fig 1. tUS device and pulse protocol.** (a) tUS transducer in housing used. The cylindrical transducer is seated into the handheld, 3D-printed housing. The signal cable enters from the bottom of the handle (not shown). A 3D tracking attachment is located on the back of the housing. (b) Illustration of a single trial of tUS. A single trial consisted of a 300-ms burst of ultrasound. The tUS device was emitting ultrasound for 36% of the time during the burst (i.e. duty cycle = 36%). The burst consisted of short, 36-ms pulses of ultrasound (i.e. pulse length = 36 ms) repeated every 1 ms (i.e pulse repetition frequency = 1000 Hz). Acoustic frequency of the ultrasound was 500 kHz. Most trials in this experiment were of the 300-ms burst protocol shown here, but a subset (25%) of tUS trials were 500-ms bursts that otherwise used the same parameters shown here (see Methods). C) Illustration of the inter-trial interval for tUS trials. Mean trial interval 10 s with ± 2 s jitter.

**TMS equipment.**  Single-trial, monophasic TMS was applied using a figure-eight coil (70 mm diameter) via a Magstim $200^2$ magnetic stimulator (Magstim, Whitland, Dyfed, UK). An infrared tracker was mounted to the TMS coil for neuronavigation. Individual resting and active motor thresholds were determined using simple adaptive PEST (SA-PEST) (Adaptive PEST TMS threshold assessment tool, Brain stimulation laboratory, Department of Psychiatry, Medical University of South Carolina). Trials had a jittered intertrial interval of 10 ± 2 s.

## EMG data analysis

**EMG post-processing.**  EMG traces were high-pass filtered with a 10-Hz cutoff (filter transition: 5–10 Hz), unless noted as unfiltered. This high-pass filter was applied to remove voltage shift and low-frequency noise. All EMG post-processing was performed using MATLAB.

**cSP measurement.**  cSPs were measured using an automated script written in MATLAB, which used a rolling standard deviation (STD) to see when EMG activity quieted below a threshold (S1 Fig). The rolling STD window had a width of 3 ms. The cSP threshold was set using the baseline EMG variability—specifically ½ STD of the rolling STD trace the 200-ms period before TMS onset. cSP onset was set to the timepoint the rolling STD first fell below threshold after the MEP peak. cSP offset was set where the rolling STD first rose back above threshold. This method is similar to other published cSP algorithms [50], albeit using rolling STD (i.e. sliding window STD) instead of the conceptually similar 'mean consecutive difference' [51].

Additional rules were necessary to handle certain edge cases the algorithm would otherwise not handle as intended. For determining cSP starts, starts were contingent on the raw EMG being near or below zero (specifically, below the same threshold value mentioned above). If no cSP onset point was found within the first 15 ms, the duration was set to zero. For determining cSP offsets, a 15-ms 'amnesty period' was included to avoid spuriously classifying MEPs as cSP offsets, since large MEPs could have high STD values in their valleys after the MEP peak. For these cases in which putative cSP duration was below 15 ms, the initially calculated offset was ignored if the EMG signal was quiet immediately following that point (i.e. below threshold 15–30 ms after cSP onset). If not, the initial threshold breach was deemed valid. Determination of cSP onsets or offsets being contingent on EMG being silent for a set period time has been used by other published cSP algorithms as well [51–54].

**MEP measurement, resting.** To quantify the size of MEPs Experiment 2, we used the area of the MEPs. The area under the curve (AUC) of the rectified MEP waveform from 20 to 120 ms after the TMS pulse was estimated via the trapezoidal method in MATLAB. AUC was not used in Experiment 1 (voluntary contractions) because of surrounding EMG activity.

**MEP measurement, voluntary contraction.** MEP peak-to-peak measurements were measured by the absolute height from the peak to the mean of the two flanking valleys of the peak. Calculation from the mean of both flanking valleys was chosen to reduce the influence of voluntary EMG activity on TMS MEP measurement—especially important in this experiment given we induced small, near-threshold MEPs. To improve accuracy of automated MEP detection during tonic contraction, MEP search was constrained using per-subject exemplar data from trials with overt MEPs. A 10-ms search window was centered around the expected MEP timepoint. Expected MEP timepoint was the median MEP timepoint during resting TMS MEP trials (Experiment 2 TMS MEP data). For the two subjects who did not participate in Experiment 2, trials with visually overt MEPs during tonic contraction were used instead (Experiment 1 TMS MEP data). This search approach was used both for the positive MEP peak and its two flanking negative valleys.

Candidate peaks and valleys in the EMG data were found using the *findpeaks* MATLAB function. To avoid minor, extraneous peaks from being selected, only peaks with a prominence and width above the 50th percentile were eligible.

TMS cSP trials were categorized into three labels: *MEP*, *stub*, and *none*. Overt MEPs (*MEP*) occurred within the 10-ms search window and had a peak-to-peak height above 0.5 mV. Potential-but-short MEPs (*stubs*) occurred within the 10-ms search window and had a peak-to-peak height below 0.5 mV. Trials with no peak that met conditions (50th percentile prominence, width) within the 10-ms search window were labeled *none* (S11 Fig).

**EMG characteristics.** Additional characteristics of the EMG traces were calculated to contrast TMS and tUS effects on the FDI EMG signal during voluntary contraction, as well compare within different periods of tUS trials. First, spectral components were determined by estimating the short-term, time-localized power spectrum of each trial and then taking the mean to get separate average spectrograms for TMS trials and tUS trials. Second, lengths of silences in the EMG signal were calculated with the same sliding window approach used to determine cSP onset and offset (see cSP measurement). Specifically, the cSP algorithm searched for a silence duration from a window centered at 0.001-second intervals from 0.05 to 0.95 s. The first and last 0.05 s were excluded to avoid edge artifacts. The results were then averaged within their respective groups to get mean silence traces.

Two additional characteristics were calculated to investigate possible EMG responses time-locked to tUS exposure: the height of the EMG (AUC) and the rate of EMG peaks. AUC was calculated as described above (MEP measurement, Resting) for two 150-ms epochs: from 200 to 50 ms before tUS onset and from 50 to 200 ms after tUS onset. This provides two 150-ms

epochs wholly covered by 'off' and 'on' periods of tUS. Rate of EMG peaks was calculated using *findpeaks* function in MATLAB on each EMG trace, binning peaks by time for the tUS-off or tUS-on periods (i.e. both the pre- and post-tUS periods were included together for tUS-off). All EMG characteristics processing was performed in MATLAB.

## Acoustic simulation

**Skull mask processing.** The same T1-weighted structural scan used for neuronavigation was used for simulations and skull thickness measurements. Specifically, we used a MPRAGE sequence (slice thickness = 0.8 mm, repetition time = 2500 ms, echo time = 3.6 ms, inversion time = 1000 ms, dimensions = 300 x 320 x 208 voxels, scanner = Siemens Prisma 3T).

For acoustic simulations and skull thickness measurements, binary skull masks were produced in BrainSuite using its "Cortical Surface Extraction Sequence" [55]. Skull masks were corrected by hand with the mask brush tool in BrainSuite. In MATLAB, skull masks were linearly interpolated to increase resolution to 0.2-mm-width voxels, and they were rotated such that the tUS trajectory was in line with the computational grid. Masks were also smoothed via morphological image processing both before and after transformation. Masks were cropped to the area of interest, creating a 484 x 484 x 484 volume. Additional skull processing details can be found in documentation for the toolbox, TUSX [56], made by these authors.

**k-Wave simulations.** Acoustic simulations were performed using k-Wave, an open-source acoustics toolbox for MATLAB [57]. Each skull mask was imported into k-Wave, providing a computational grid spacing of 0.2 mm. To simulate the transducer, we set a curved disc pressure source (k-Wave function: *makeBowl*) with a curvature radius of 30 mm and aperture of 30 mm to mirror the focal length and width of the real transducer, respectively. The pressure source emitted a 0.5 MHz sine wave, resulting in a grid points per wavelength (PPW) of 14.8. Simulations were performed at a temporal interval of 285 temporal points per period (PPP) for a Courant-Friedreichs-Lewy (CFL) number of 0.0519. Perfectly matched layers (PML) of 14 grid points were added for a total grid size of 512 x 512 x 512.

To allow comparison to real-world pressure measurements in the water tank, each tUS trajectory was simulated twice: once to simulate propagation through the skull and once to simulate propagation through water. For skull simulations, the same acoustic properties were given for all points within the skull mask: density of 1732 kg/m$^3$, a sound speed of 2850 m/s, and an alpha coefficient of 8.83 [dB/(MHz$^y$ cm)] [58]. The use of homogenous skull acoustic properties has been shown to be effective in simulations within the frequencies used here [59–61]. All values not within the skull mask were given bulk acoustic values of brain: 1546.3 kg/m$^3$, a sound speed of 1035 m/s, and an alpha coefficient of 0.646 [dB/(MHz$^y$ cm)] [62]. Homogenous water simulations were given acoustic properties of water at 20˚C: a density of 998 kg/m$^3$, a sound speed of 1482 m/s, and an attenuation constant of $2.88 \times 10^{-4}$ [Np / m] [62]. An alpha power of 1.43 was used for all simulations.

To estimate in-brain pressures experienced by participants for a given tUS trajectory, we used a ratio of pressures from the skull and water simulations. The estimated pressure ($P_{est.}$) at a given location was calculated as

$$P_{est.} = \frac{P_{skull\ sim.}}{P_{water\ sim.}} \times P_{water,real}$$

Where $P_{skull\ sim.}$ is the temporal maximum pressure value at that same location in the skull simulation for the specific subject and trajectory. $P_{water\ sim.}$ is the temporal maximum pressure value at the focal point of the water simulation. $P_{water,real}$ is the temporal maximum pressure value at the focal point measured in a water tank of degassed water using the same parameters

as used in the experiment (see tUS equipment). To avoid any potential outliers in the simulated data, spatial averaging was performed on $P_{skull\ sim.}$ and $P_{water\ sim.}$ by taking the mean within a 0.6-mm radius sphere. $P_{water,real}$ was 1.40 MPa for all subjects except one (sbj11), whose $P_{water,real}$ was 1.13 MPa due to the lower waveform generator setting used for that session (user error).

Simulations were performed on the Ahmanson-Lovelace Brain Mapping Center computational cluster. Each simulation instance was allocated 24 CPU cores and took approximately 2.5 hours with the C++ implementation of k-Wave (kspaceFirstOrder3D-OMP) [63].

**Target registration.** NIBS targets and the location of $M1_{hand}$ were determined via registration to standardized stereotactic space (Montreal Neurological Institute, MNI). Registration was performed with FSL's FNIRT/FLIRT tools [64, 65]. $M1_{hand}$ was set to the voxel closest to the MNI coordinates x = −39, y = −24, z = 57 [43].

**Exposure.** An estimate of cumulative $M1_{hand}$ exposure was made by multiplying the individual peak pressure at the $M1_{hand}$ voxel for each of the three tUS trajectories by the time the tUS device was on for that location. Specifically, exposure was defined as

$$\sum_{traj=1}^{n} P_{traj} \times Time_{traj}$$

for **$n$** number of tUS trajectories, where **$P_{traj}$** is the pressure at the $M1_{hand}$ voxel for that trajectory, and **$Time_{traj}$** is time tUS was on for that trajectory. We display these values in the form Pascal-hours (Pa·hr).

## Statistics

**Experiment 1, cSPs.** TMS cSP durations vs. aMT was analyzed with a one-way repeated-measures ANOVA. For post-hoc tests, Welch's t-tests were performed on the group distribution pairs (90 & 100, 90 & 110, 100 & 110) using the cSP durations demeaned to their respective subject mean (*Duration–Subject Mean*).

Rate of EMG peaks on vs. off during tUS was analyzed using a paired t-test, with the rates 'on' rate and the 'off' rate for each trial paired. In other words, the tUS-off portions of the 1000-ms epochs served as the control.

**Experiment 2, Cortical excitability.** $M1_{hand}$ excitability was analyzed with a paired ranked non-parametric t-test since the distribution was non-gaussian. We performed this using resampling using a script in R [66]. For null hypothesis testing, permutation was used to create a null distribution of all permutations of before- and after-tUS values swapped within subjects (256 permutations). All 16 medians of each permutation (8 subjects, 2 conditions) were then ranked against one another. The difference of the means of the permuted group ranks for each permutation was used as the values of the null distribution. The value of p equaled the number of permutations in which the absolute difference of the mean ranks was greater than the real absolute difference of the mean ranks.

Confidence intervals were calculated using the bootstrap method (10,000 bootstrap samples). Each bootstrap sample was made by sampling with replacement the 16 real median MEP sizes (8 subjects, 2 conditions). The 95% confidence intervals were set as the 2.5th and 97.5th percentiles of the bootstrap samples' differences of the group means.

To investigate any association between cortical excitability change and total tUS exposure of $M1_{hand}$, an estimate of total tUS exposure for a participant was calculated with the following

formula:

$$\sum_{traj=1}^{n} P_{traj} \times Time_{traj}$$

Where **n** is the number of tUS trajectories used with that participant, **$P_{traj}$** is the pressure (estimated) at $M1_{hand}$ voxel for that trajectory, and **$Time_{traj}$** is the time the tUS device was on for that trajectory. Determination of **$P_{traj}$** is outlined in k-Wave simulations. The spearman correlation coefficient ($r_s$) was calculated for these values.

**cSP null distribution.** To bootstrap a null distribution of cSP lengths our automated cSP algorithm would find if applied to null, non-cSP data, we used a sliding window approach on non-cSP data. This data was real EMG traces collected during tonic contraction by the same subjects and sessions as Experiments 1 and 2—specifically, the one-second tonic contraction trials collected during tUS exposure. tUS trials were deemed valid as null EMG traces since we saw no change in EMG traces between tUS on vs. tUS off (Figs 2 and 3, S2–S4 Figs). For every null trial, the cSP algorithm searched for a silence duration from a window centered from each 0.001-second interval from 0.05 to 0.95 s. The first and last 0.05 s were excluded to avoid edge artifacts. All trials, subjects, and sliding-window increments were grouped into a single distribution (686,457 sliding window samples).

## Results

### Experiment 1

**TMS cSPs.** Single-pulse TMS was performed over left $M1_{hand}$ during tonic contraction of the FDI muscle (n = 10). TMS was delivered at 90%, 100%, and 110% aMT. cSP duration

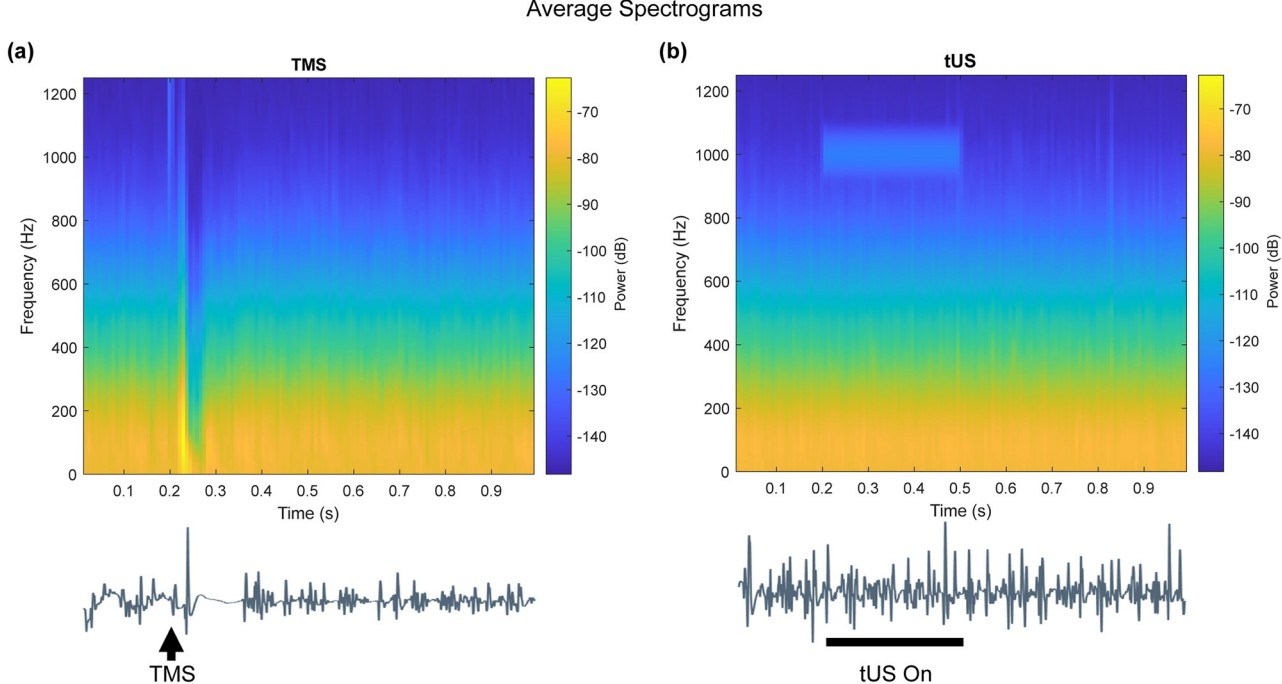

**Fig 2. Average spectrograms of EMG during tonic contraction.** (a) TMS trials. (b) tUS trials (300-ms tUS duration trials only). Signal around the 1000-Hz range from 0.2–0.5 s during tUS trials is noise recorded from the amplifier. This frequency component matches the pulse repetition frequency. Example EMG traces placed below the spectrograms illustrate timing.

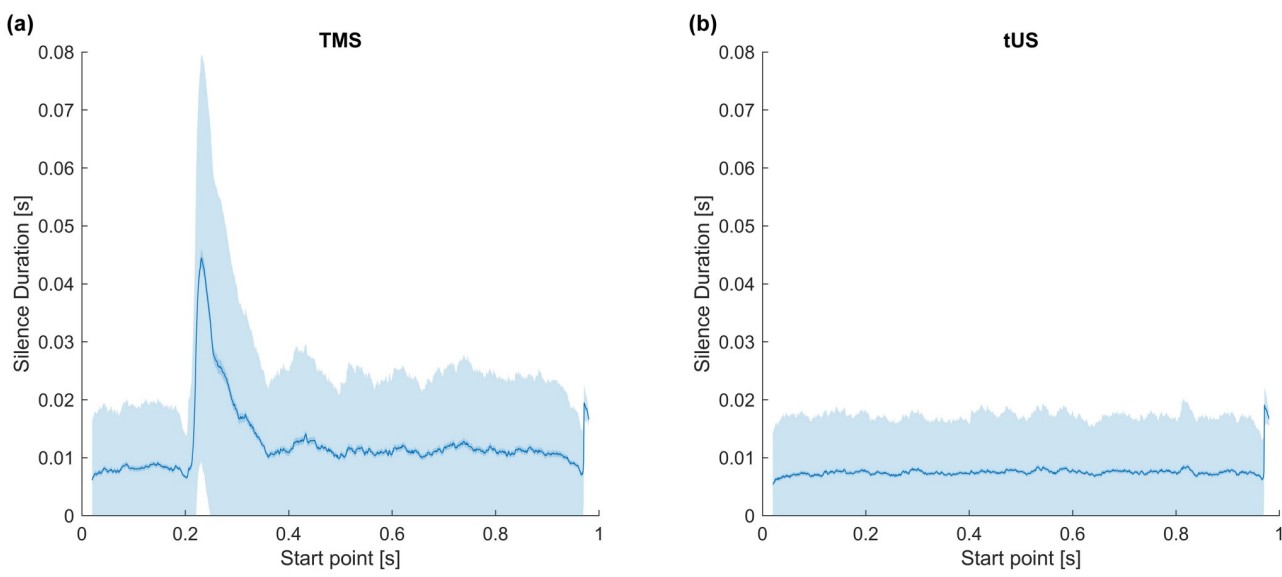

**Fig 3. Silence durations across trials, using a sliding window.** Duration is as measured from the start point of a given sliding window iteration. (a) TMS. (b) tUS. X-axis: Timepoint measured from. Middle Trace: Mean silence duration. Inner Margin: SEM. Outer Margin: STD.

increased with TMS intensity (ANOVA: $F_{2,16}$: 26.31, p < 0.001; Welch's t-tests: p < 0.001, all pairs) (Fig 4, S9 Fig). The aMT threshold for one subject (sbj11) was set mistakenly low, resulting in TMS intensities lower than intended and therefore elicited very few cSPs.

We also examined the size and presence of MEPs preceding cSPs. For trials with overt MEPs, the lengths of the subsequent silent periods were noticeably longer than would be seen by chance (S5 Fig). Among trials with a peak within the expected 10-ms time window, trials with a peak smaller than the standard MEP peak-to-peak amplitude threshold of 0.5-mV, henceforth referred to as "stub" trials (S10 Fig), mostly showed silences within lengths that would be seen by chance (S5 Fig). However, some "stub" trials did show long silence durations on par with those of overt MEP cSPs. Lastly, trials in which there was *no* peak within the 10-ms time window showed silence durations within what would be seen by chance, with only one of these trials showing a silence above the 95th percentile of the null distribution.

**tUS.** Single-burst tUS was performed over left M1$_{hand}$ during tonic contraction of the FDI muscle (n = 10). The 300-ms or 500-ms bursts were delivered at three trajectories per subject, one of which was also the trajectory for TMS.

No overt silent periods were visible during single trials of tUS stimulation. To investigate whether tUS caused any suppression of the EMG trace, we investigated the height of the EMG traces (area-under-the-curve, AUC) (S2 Fig), the lengths of the intermittent contraction silences, the rate of EMG peaks (S3 Fig), and the spectral components of the EMG signals. While a drop in signal power of the spectral components occurs due to TMS-induced cSP, no spectral changes are visible during tUS trials (Fig 2). The same disparity is seen comparing the length of silences in the EMG signal, with a clear rise in mean silence period in response to TMS-induced cSP but no change in response to tUS (Fig 3).

For comparisons performed within tUS trials, the height of the EMG traces showed no difference directly before versus after tUS onset (150-ms epochs before vs. after tUS) (S2 Fig). The rate of EMG peaks while tUS was on vs. off also showed no significant difference (S3 and S4 Figs), with a paired t-test confirming there was only a small but statistically insignificant tUS effect on rate of EMG peaks (Delta: -0.91 Hz; 95% CI: -1.99, 0.16 Hz; p = 0.095) (S7 Fig).

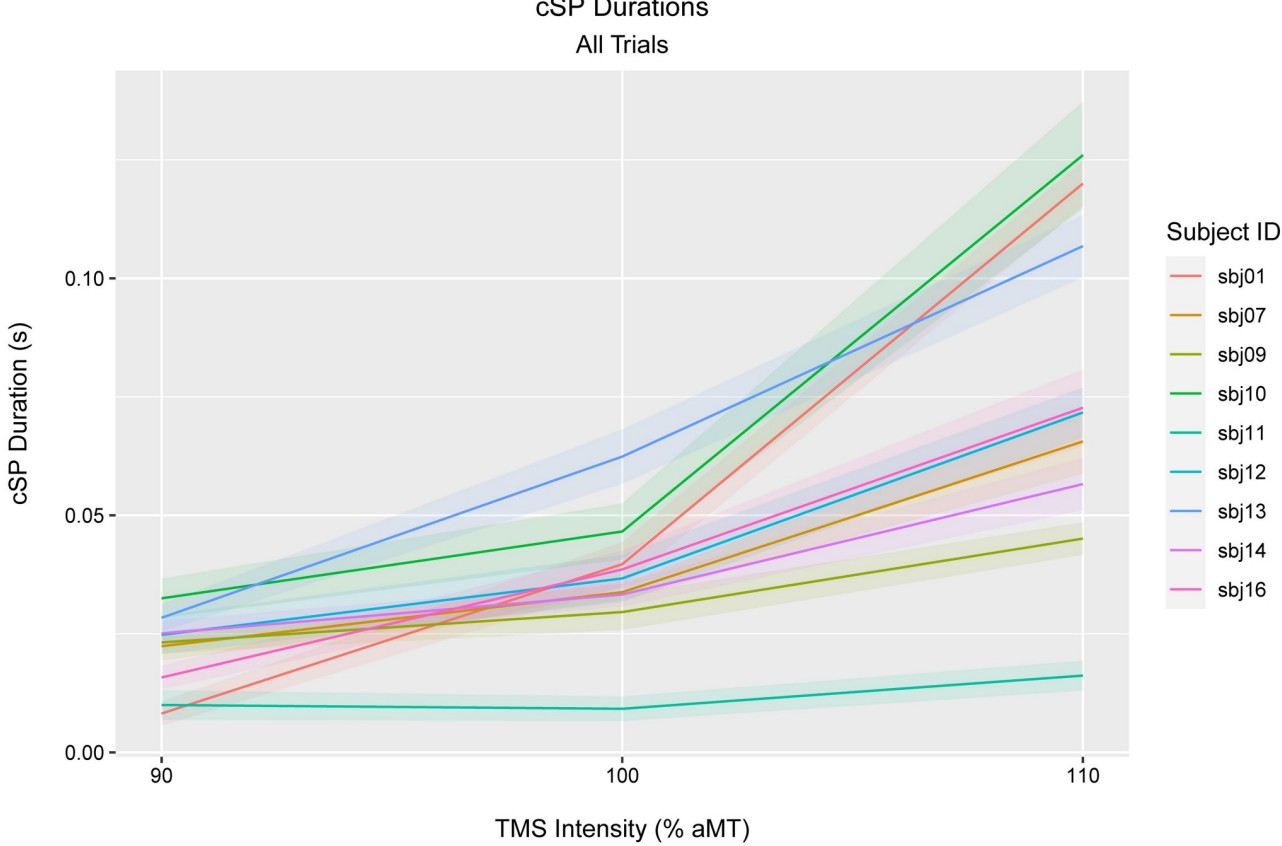

**Fig 4. cSP length for each TMS intensity, per subject.** Line: median. Ribbon: standard error of the mean. One subject (sbj08) with whom resting motor threshold was used is not shown here (see S8 Fig).

### Experiment 2

**Cortical excitability.** Cortical excitability was gauged before and after exposure to tUS by recording MEPs from single-pulse TMS over $M1_{hand}$ (n = 8). Both the pre-tUS and post-tUS measurements (1-min post-tUS) consisted of 20 suprathreshold TMS trials with an intertrial interval of 10 ± 2 s. The size of TMS-induced MEPs did not vary between before- and after-tUS conditions, according to a ranked paired non-parametric t-test (Fig 5) (Delta: -0.64 mV-ms; 95% CI: -2.39, 0.84 mV-ms; p = 0.51).

**Exposure & excitability.** To investigate the variability that was present among cortical excitability responses, we compared subjects' cortical excitability change to total estimated tUS exposure in the session. tUS exposure estimates were made using acoustic simulations in models that matched each experimental setup, with skull data computed from structural MRI of the tUS participants. These data showed no obvious correlation between $M1_{hand}$ exposure and cortical excitability change (n = 8), with a spearman correlation coefficient of -0.21 (Fig 6).

### Acoustic simulation

Acoustic simulation results suggest we very accurately 'hit' targets we were aiming at (Fig 7). tUS produced pressures in an ellipsoid focus, with a mean FWHM with of 4.5 mm (S1 Table).

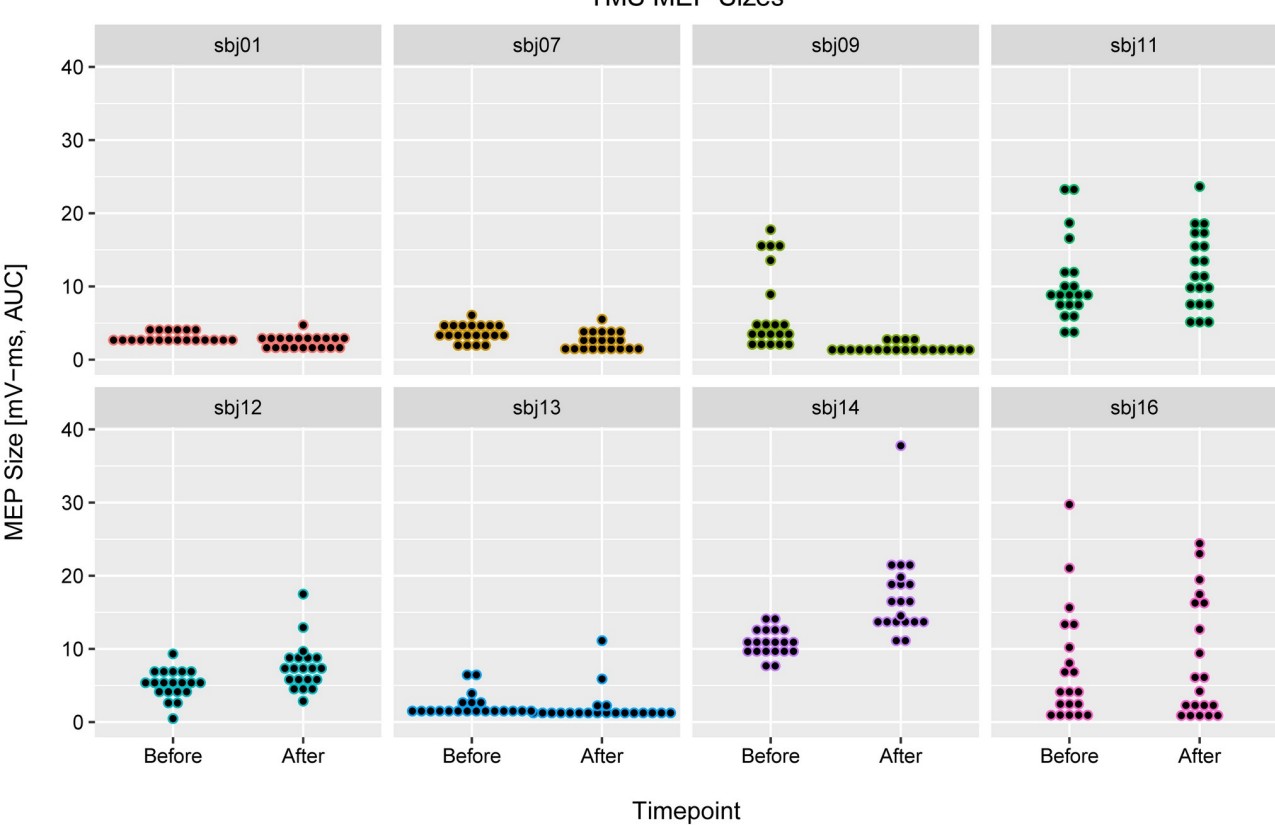

**Fig 5. TMS MEP sizes, before and after tUS.** Each subplot contains data from one subject: trials before and after tUS on the left and right, respectively. Each point marks the size of one MEP (area under the curve, mV-ms). Points are organized into vertical-axis bins to allow visualization akin to a violin plot (i.e. original data are not discrete values). Using amplitude instead of area under the curve did not change the results (S15 Fig).

## Discussion

### Experiment 1

**No tUS-MEPs.**   We were unable to elicit tUS-induced MEPs at safe intensities. This was the expected outcome, given the lack of MEPs in previous human tUS studies [31, 32, 67]. Some recent animal model work suggests that motor activation via ultrasound stimulation of motor cortex may not be feasible, proposing that previously reported motor contractions in anesthetized animals possibly relied on auditory mechanisms [68, 69]; however, others suggests motor activation via ultrasound stimulation may still be possible [70]. We did not investigate for potential resting tUS-MEPs during rest beyond a single pilot subject. No tUS-induced MEPs appeared during active contraction trials either.

**Cortical silent period, TMS.**   Our data found no cSPs that occurred without a preceding TMS-evoked MEP (S5 Fig), despite structuring the study to facilitate a high prevalence of near-threshold MEPs. As such, this contradicts claims in the literature that TMS-induced cSPs can occur without an MEP [29, 30, 50]. One explanation for this difference is that a small "stub" MEP could precede reported "MEP-less" cSPs, with its amplitude not surpassing the amplitude of tonic contraction. This is supported in these data by the consistent appearance of an EMG peak within the latency window expected for TMS-evoked MEPs.

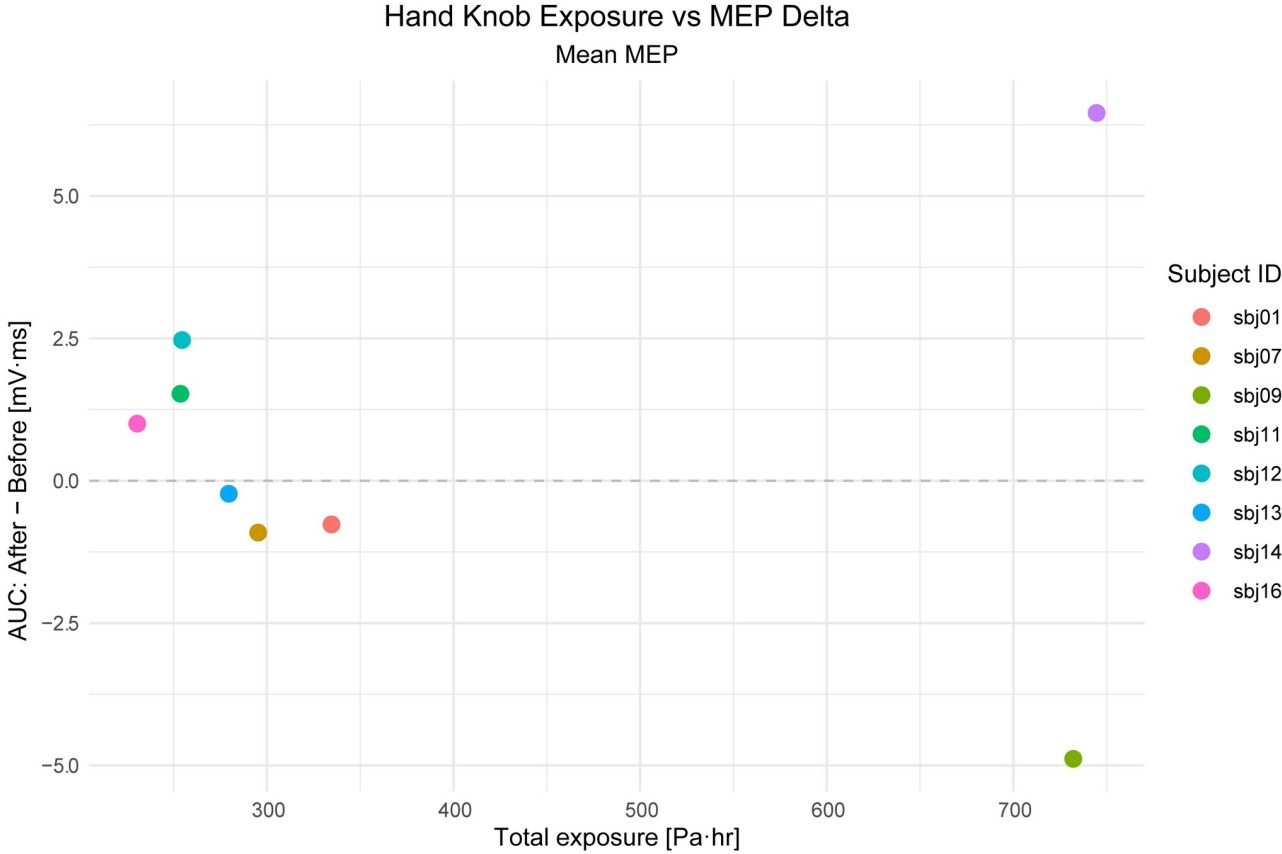

**Fig 6. M1$_{hand}$ exposure vs. cortical excitability.** Estimate of total exposure of M1$_{hand}$ to tUS cumulatively during the session (horizontal axis) compared to the change in cortical excitability, as measured by TMS-evoked MEP (vertical axis). $R_s$ = -0.21.

If this MEP-cSP dependency is true, this could suggest that cortical silent period is dependent on the recruitment of M1 motor units. TMS preferentially depolarizes axons [71–74]. While still early in investigation, tUS conversely has often been proposed to preferentially affect inhibitory neurons [2, 35–38]—though this is certainly not universal [75]. If these assumptions are true, this could explain why tUS struggled to silence corticospinal output in these data.

To be clear regarding the notion of "stub" trials within our TMS data, we do not believe all TMS trials classified as a "stub" by the algorithm are MEPs. Rather, we believe there are two underlying distributions that fall under the "stub" designation. The first: trials in which there is a TMS-evoked MEP that is shorter than the standard threshold (0.5 mV). The second: trials in which there was an EMG peak produced *by chance*—created by a peak in tonic muscle EMG activity that fell within the expected time window (S10 Fig).

We must also note: For cSP length determination, we took a conservative approach on brief EMG activity flanked by periods of silence, referred to in the literature as late excitatory potentials (LEPs) [76–78]. Of the two silent periods flanking an LEP, we included only the first silent period when measuring cSP duration. Since these LEPs appear heuristically as short EMG disruptions of a longer cSP, a visual inspection of the data suggests that ignoring these LEPs would have resulted in less variable cSP durations within blocks (S6 Fig). For comparison, these LEPs have at times been ignored in past by-hand cSP measurements [50].

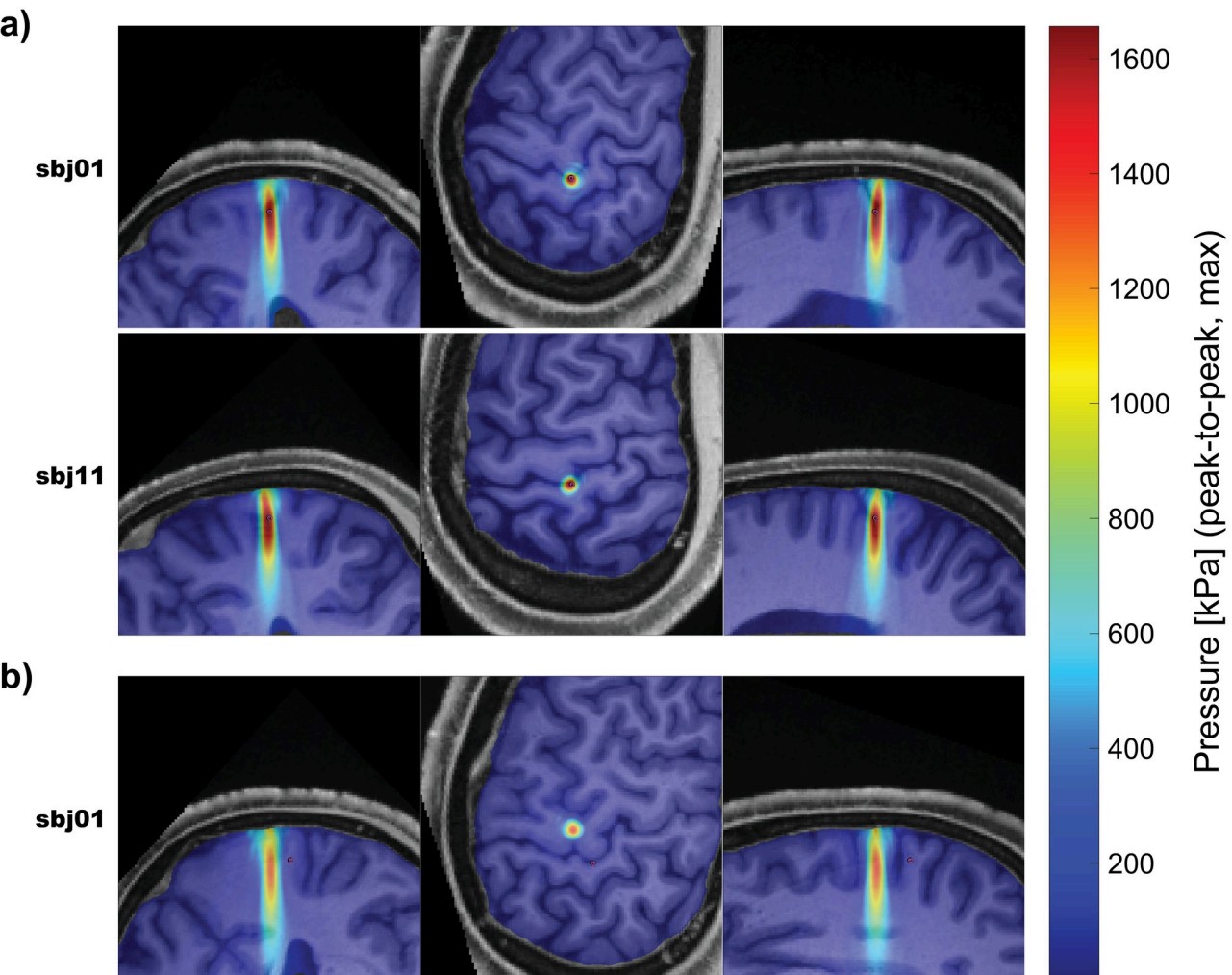

**Fig 7. Simulated pressures (examples).** Examples of simulated pressures for targets used in the experiment. These three examples were chosen to highlight how a small spatial deviation can significantly affect sonication exposure at a given coordinate. Simulated pressure maps are overlayed over the respective structural MRI. One example per row; all three dimensions per row. Slices shown were made at the maximum pressure value. Note that these are not standard slices (i.e. coronal, sagittal, horizontal), since the volume was reoriented as part of pre-simulation processing, A small magenta circle denotes the registered M1 coordinate. (a) Two example trajectories that were aimed at the respective subject's M1$_{hand}$. (b) One example trajectory that was not aimed at the respective subject's M1$_{hand}$. Pressure maps for all subjects and trajectories are available in the (S14 Fig).

**Cortical silent period, tUS.** Single-burst tUS of the hand area in the left motor cortex did not affect tonic muscle contraction of the FDI muscle. Specifically, there were no deviations in gap duration between tonic muscle spikes (Fig 3), spectral components (Fig 2), or prevalence of EMG peaks (S3 and S4 Figs). This is in sharp contrast to the lengthy silent periods from single-pulse TMS. This is also in contrast to a growing number of studies showing inhibitory effects by tUS [2, 31–33, 35–38].

While these data revealed no time-locked tUS effects, recent work using spatially overlapping TMS and tUS devices have found measurable effects in certain conditions by using TMS as an instantaneous probe. Specifically, TMS-induced MEPs were suppressed by concurrent tUS bursts using spatially targeting the same area of M1 [31–33]. Among these studies, *Fomenko, Chen et al., 2020* showed a unique finding that provides insight into our results:

while tUS had robust suppression of MEPs amplitudes evoked while the subject's hand muscles were at rest (i.e. resting MEP), tUS had no effect on MEP amplitudes evoked while the subject's hand muscles were contracting (i.e. active MEP). In other words, the presence of the target muscle tonically contracting was tied to the loss of the suppressive effects of tUS. If ongoing voluntary motor activity precludes, or at least greatly impedes, measurable tUS effects in M1, this could explain why we failed to find time-locked effects of tUS on the EMG signal.

This leaves the question of why voluntary motor activity would prevent tUS effects. One hypothesis we propose is that there could be a floor or ceiling effect involving inhibitory interneurons, which have been proposed to be preferentially affected by tUS [2, 32, 35–38]. Indeed, electrophysiology evidence [79] from optogenetic tagging [80] shows at least some M1 interneurons increase activity during voluntary motor behavior. Therefore, some of the cells that tUS would have preferentially affected (inhibitory interneurons, [2, 32, 35–38]) are already being actively modulated at the time tUS energies enter the tissue, thus making further modulation of ongoing activity more difficult. Assuming tUS acts on neurons by opening ion channels [81–83], we hypothesize this difficulty in modulation could emerge due to the interneurons in question being in a state of high membrane conductance at the time of tUS (i.e. their tUS-sensitive ion channels are already open).

Along with tUS not affecting active MEP, the cSPs that follow tUS-targeted active MEPs were not affected either [32]. This too is interesting considering the cSP phenomenon is dependent on cortical interneurons [26–28]. This matches our results in that tUS did not increase periods of EMG silence. That said, our approaches do differ. *Fomenko, Chen et al.* used TMS to instigate an MEP while we did not. Additionally, the approach to tUS timing differed. In *Fomenko, Chen et al.*, tUS was turned off 10 ms after TMS onset (mean cSP length = 138 ms, *Fomenko, Chen et al.*), while sonication overlapped the full target epoch in our approach. Having noted the differences, the fact that both studies showed no change in EMG activity when the target muscle was being voluntarily contracted is quite intriguing.

Another factor that could be at play is a theory proposed by *Fomenko, Chen, et al.*: that tUS may preferentially modulate $GABA_A$ activity relative to that of $GABA_B$. Their proposal is based on the fact that tUS increased short-interval intracortical inhibition (SICI) in their study, which is considered a $GABA_A$-mediated phenomenon [84]. Conversely, they also did not see changes in $GABA_B$-mediated effects tested [32]: the paired-pulse paradigm long-interval intracortical inhibition [85, 86] and cSP [26]. Considering that our tUS during tonic contraction paradigm is indeed similar to a cSP paradigm of TMS during tonic contraction, it is possible that our paradigm would be sensitive to $GABA_B$-related activity levels. However, it is the *late* part of the cSP that is believed to be $GABA_B$-mediated [26], and we did not even see short increases of EMG silence. As such, we are not sure whether or not our data would add to the $GABA_B$ discussion regarding tUS. But, overall the notion of heterogeneity of tUS sensitivity among $GABA_A$ channels and $GABA_B$-linked channels is certainly intriguing and warrants further investigation—potentially in combination with the notion of whether tonic activity impedes the ability of tUS to affect M1 circuitry.

Though we were looking for effects on EMG signal, we know from the TMS literature that it *is* possible to induce detectable excitation of motor cortex without a measurable peripheral effect. Specifically, electroencephalography (EEG) recordings during subthreshold single-pulse TMS show significant TMS-evoked potentials, despite a lack of a peripheral MEP [87]. Given this, it is possible there were tUS effects that did not interact with corticospinal projections— making them undetectable by EMG and therefore undetectable by our experiment. While EEG is not necessarily a more sensitive readout to study all mechanisms, such as for certain corticospinal excitability experiments [88], adding methods like EEG and fMRI that record

cortical effects directly could provide a more complete picture of time-locked tUS effects in future investigations.

## Experiment 2

**Cortical excitability.**   Our data showed no group difference in M1 excitability in response to tUS, as indicated by no change in MEPs evoked by TMS at the same M1 trajectory. Our data differ from data by Gibson and colleagues that showed increased excitability of M1 after ultrasound exposure [34]. There are study design differences that could have driven this disparity. The first is the tUS device used. While we used a 500-kHz single-element focused transducer, *Gibson et. Al 2018* used an imaging ultrasound device, which consisted of an array of 80 transducer elements emitting frequencies in a range between 1.53 and 3.13 MHz. This frequency range is noteworthy because acoustic attenuation increases as a function of acoustic frequency [89–91]. *Gibson et al. 2018* cited papers that used ultrasound imaging devices to image through the skull as evidence of the device's validity for use over M1. However, all studies they cited placed the device over the temporal window—an area of the skull that is significantly thinner than that over M1 (temporal window: ~3 mm; parietal bone: ~6 mm) [92, 93]. In fact, measurement of ultrasound propagation through ex-vivo human parietal skull shows that little to no energy is transmitted at frequencies above ~1.5 MHz [90, 94–96].

But importantly, our stimulation protocol may not have been ideal for inducing offline effects. We delivered separate bursts of ultrasound (duration: 300–500 ms each) with long gaps between bursts (8–12 s inter-burst interval). Our ~10-second inter-burst interval is very slow compared to repetitive TMS protocols used to modulate cortical excitability, and it ventures into the intertrial interval range suggested for use to avoid central habituation effect in sensory stimulation studies [97–99]. It is also starkly different from *Gibson et al.*, which delivered constant exposure to an ultrasound imaging protocol for 2 minutes [34]. Acquisition of Experiment 1 and Experiment 2 data were integrated, i.e. the same tUS trials were used to both investigate time-locked (Experiment 1) and offline effects (Experiment 2), so changing the tUS protocol was not an option for this study. As such, future tUS studies may want to use more compressed protocols with shorter inter-burst intervals, if neuromodulation is the aim. A compressed approach was shown to successfully affect tUS targets in non-human primates, as shown by the reduction of resting-state fMRI connectivity following a 40-s tUS protocol (pulse repetition frequency: 10 Hz; pulse length: 30 ms) [100, 101]. These non-human primate studies showed effects lasting up to two hours after stimulation. However, they also used tUS intensities, 24.1–31.7 W/cm$^2$, that were significantly higher than the levels used here or any other human tUS study (human max.: 4.9 W/cm$^2$ in *Legon et al. 2014*). While histological examination in these studies revealed no microstructure damage, the protocol in question still corresponds to a mechanical index of ~3.6—higher than the 1.9 maximum allowed by the FDA for diagnostic imaging [91, 102]. Given that the most robust tUS effects seem to occur at high intensity levels, future studies will need to carefully explore whether consistent, behaviorally relevant tUS effects are feasible at intensities safe for human exposure. Replication of repetitive tUS protocols, at lower intensities, will likely be the first step. Indeed, recently published work showed an increase in motor excitability in humans up to 30 minutes following a condensed tUS protocol (pulse repetition frequency: 5 Hz; burst duration: 500 ms; inter-burst interval: 1.1 s) at intensities safe for human use [103]. Considering the device used by *Zeng et al.* was very similar to the one used here, this reinforces our interpretation that the stimulation protocol used here was not ideal for inducing offline effects.

**Exposure & excitability.**   With this small sample size (n = 8), we saw no correlation between M1$_{hand}$ exposure and cortical excitability change, though the calculation of this

correlation was also low-powered. This conclusion is to be expected since this study was not designed to investigate such a correlation, with M1$_{hand}$ exposure effectively stratified into two levels depending on whether the M1$_{hand}$ target was used once or twice (i.e. whether it was the primary NIBS target, "TMS target"). Studies that wish to investigate potential correlation effects of exposure levels would need to expose participants at a variety of different levels and have a larger sample size than used here to increase statistical power compared.

While we performed acoustic simulation on 907 cm$^3$ volumes (~300–350 cm$^3$ of which were grey or white matter), we still chose to tabulate cumulative exposure for a single location: M1$_{hand}$. M1$_{hand}$ was chosen because it is the most reasonable small-volume, easily identifiable cortical area that is known to play a direct role in hand muscle contraction. It is, however, possible that directly targeting M1$_{hand}$ may not be the ideal choice for modulating voluntary muscle contraction with tUS. This notion of placement is especially important given the spatially precise nature of tUS compared to a TMS or transcranial electrical stimulation—especially with a small, focused ultrasound transducer as used here.

### tUS and M1

While there could be multiple causes, one potential explanation could lie in cytoarchitectural differences between brain regions. Crucially, motor cortex has significantly lower neuronal density compared to somatosensory and visual cortex [104–107]. As such, for a tUS pressure field of a given size, the number of individual neurons that fall within the focus would be lower in M1 compared to primary somatosensory (S1) or primary visual cortex (V1). This disparity could leave M1 neurons at a relative disadvantage for reaching thresholds to create detectable systems-level effects from tUS exposure. Additionally, the inherent cytoarchitectural and circuitry differences between 'output' cortical regions, like M1, compared to 'input' cortical regions, like S1 and V1, could likely have a significant role.

Alternatively, the fact that we performed tUS during ongoing motor activity may have been a factor, since recent data showed that tUS-mediated suppression of TMS MEPs [31, 33] was not present during tonic motor contraction of the target muscle [32]. Assuming the theory that tUS acts on neurons by opening ion channels [81–83], ultrasonic opening of channels should be less impactful if those same channels are already in a state of high conductance.

### Conclusion

We performed neuronavigated tUS and TMS of primary motor cortex (M1) in healthy volunteers. We found no concurrent change in finger EMG activity from tUS of M1 during voluntary muscle contraction. We also did not find any consistent effect of tUS M1 exposure on motor cortex excitability, as measured by single-pulse TMS of M1. We performed acoustic simulations using structural MRI of the study participants to estimate the degree and location of ultrasound intracranially. Using these simulations, we were unable to find any correlation between cumulative ultrasound exposure of the M1 hand area and M1 excitability change with the uncondensed tUS protocol used here.

Within the TMS-only data, our data suggest that cortical silent periods (cSP) may be contingent on a motor evoked potential (MEP) occurring at cSP onset, though at times the MEP may elude visual detection due to a small amplitude that does not rise above the level of tonic muscle activity. This finding questions previous reports of cSPs without MEPs [29, 30, 50].

While the negative tUS results reported here mirror struggles some other investigators have shown when attempting to elicit measurable modulation of M1 by tUS, this was also a pilot study with small sample sizes (n = 8; n = 10). As such, clearer results may emerge with larger datasets or changes in methodology [103].

Lastly, given that both we and others [32] failed to show tUS effects during tonic motor contraction, futures investigation may be warranted to consider how the state of channel conductance may affect tUS outcomes.

## Supporting information

**S1 Fig. Visualization of the automated cSP detection method.** Top) Sliding window standard deviation trace of a single trial EMG trace. The black horizontal line marks the detection threshold. The vertical green lines mark the beginning and end of the detected cSP. Bottom) The original high-pass filtered EMG trace.
(PDF)

**S2 Fig. Area-under-the-curve of tUS traces.** Level of EMG activity during different sections of tUS trials. Left) Baseline, -200 to -50 ms before onset. Right) 0 to 150 ms after tUS onset (first 150 ms of tUS exposure). Values were normalized via dividing by the trial mean.
(PDF)

**S3 Fig. Prevalence of EMG peaks.** Rate of EMG peaks during tonic contraction per subject. Left) TMS trials. The drop in EMG activity due to TMS-induced cSPs is visible soon after TMS onset at 0.2 s. Right) tUS trials. tUS on from 0.2–0.5 s A sliding window approach (1-ms steps) checked if an EMG peak occurred during a 5-ms time window following that point. Peaks were detected using the 'findpeaks' function in MATLAB. Each row contains data for all trials per subject. Color shows percentage of trials that had a peak during that time window. Percentage data was smoothed with a moving mean (~6-ms window). EMG traces were high pass filtered at 10 Hz.
(PDF)

**S4 Fig. Rate of EMG peaks during a single tUS trial.** One dot per trial per condition ('Off' and 'On'). EMG traces were bandpass filtered to 10–800 Hz.
(PDF)

**S5 Fig. Silence null distribution bootstrap.** (Top) Distribution of silence lengths when cSP length algorithm is run from different time points along each EMG from a contracting finger. Sliding window approach was used to bootstrap these values, with a sliding window step size of 0.001 seconds. Histogram bin width 0.001 seconds. "Null" data were the one-second tonic contraction trials during tUS exposure. tUS trials were deemed valid as null EMG traces since we saw no change in EMG traces between tUS on vs. tUS off (Figs 2 and 3, S4 Fig). The first and last 50 ms were removed to avoid boundary effects. 686,457 sliding window samples (Bottom) Lengths of silent periods for trials grouped into three categories: a clear MEP was present ("MEP", blue), a small peak that may have been an MEP was present ("Stub", green), and no detectable peak was present ("None", red). Null distribution (see Top) overlaid in orange. MEP: 361. Stub: 151. None: 18.
(PDF)

**S6 Fig. Examples of cSPs with late excitatory potentials (LEPs).** X-axis: Time [s]. TMS onset at 0 s. Examples are from multiple subjects.
(PDF)

**S7 Fig. Distributions of rate of EMG peaks during a single tUS trial.** All subjects; all trials. EMG traces were bandpass filtered to 10–800 Hz. Same data as shown per subject in S4 Fig.

Difference of the mean rates of EMG peaks were only marginally lower for tUS 'On' vs. tUS 'Off' (Delta: -0.91 Hz; 95% CI: -1.99, 0.16 Hz; p = 0.095; paired t-test).
(PDF)

**S8 Fig. Violin plots of TMS-evoked cSP durations separated by research participant ("Subject ID").** Same data as shown in Fig 4. Sbj08 subject was excluded from Fig 4 since their cSP trials were not used in analysis (due to use of different TMS levels). Non-zero silence durations were recorded for 90% aMT for two reasons. First, there are inherent gaps between EMG peaks during tonic contraction, which average ~7.5 ms with our algorithm and our data (S5 Fig, Top). Second, TMS of M1 results in a distribution of responses, with some trials reaching MEP and cSP threshold while other trials do not (i.e. motor thresholds are never hard cutoffs).
(PDF)

**S9 Fig. cSP durations demeaned by subject mean.** Histograms and density plots shown by % aMT. Welch's t-tests performed as post-hoc tests confirmed cSP duration increased by % aMT ($p < 0.001$, all pairs). For non-demeaned data see Fig 5 and S8 Fig. One subject (sbj08) with whom resting motor threshold was used is not shown here (see S8 Fig).
(PDF)

**S10 Fig. Comparison of three different results from single-pulse TMS during tonic contraction.** All three trials are from the same subject. The automated trial designation classified each trial (**Top**, **Middle**, **Bottom**) as:"cSP","Stub","Stub". These two distinct examples of scenarios that fall under the "Stub" designation, as determined by the algorithm. This illustrates that there are likely two main distributions of trials that fall under the "Stub" designation. The first: trials in which there is a TMS-evoked MEP that is shorter than the standard threshold (0.5 mV). The second: trials in which there was by chance an EMG peak produced by tonic muscle contraction that fell within the expected time window.
(PDF)

**S11 Fig. All tonic contraction TMS trials designated as "None".** TMS onset at 0 s. "None" trials had no prominent EMG peak within the 10-ms search window. Peaks had to be above the $50^{th}$ percentile for peak prominence and above the $50^{th}$ percentile for peak width (for EMG peaks within the 1-second trial).
(PDF)

**S12 Fig. TMS search grid and trajectories.** Illustration of the TMS search grid used in both 2D and 3D. The grid's origin (white) was placed at MNI coordinates that correspond to $M1_{hand}$ as based on a meta-analysis of fMRI motor experiments: x = −39, y = −24, z = 57 [43]. The other five targets on the grid (grey) were in a 12 voxel-width grid (9.6 mm grid interval) around $M1_{hand}$ in subject space. See EMG and NIBS Placement.
(PDF)

**S13 Fig. TMS MEP waveforms, cortical excitability.** MEP waveforms for Experiment 2 for each subject. MEPs measured before (blue) and after (green) tUS exposure. (a) All waveforms. (b) Averaged waveforms.
(TIF)

**S14 Fig. Pressure maps, all subjects.** Simulated pressure maps are overlayed over the respective structural MRI. One file per subject. One target per row. 3 slices per target. Slices shown at the maximum pressure value. Note: these are not standard slices (i.e. coronal, sagittal,

horizontal), since the volume was reoriented as part of pre-simulation processing, A small magenta circle denotes the registered M1 coordinate.
(ZIP)

**S15 Fig. TMS MEP amplitudes, before and after tUS.** Each subplot contains data from one subject: trials before and after tUS on the left and right, respectively. Each point marks one MEP amplitude (mV). Points are organized into vertical-axis bins to aid in visualization. Delta: 0.1 mV; 95% CI: -0.32, 0.29 mV; p = 0.17. No significant difference was found when using area under the curve, as well (Fig 5).
(TIF)

**S16 Fig. Transducer and skull mask location, example.** A 3D render of the volumetric data from one example simulation to show the position of the transducer (red) relative to the skull (grey). Same render viewed from (a) the side and (b) the top down. Images show that neither the transducer nor critical portions of the skull mask were cut off by cropping done during preprocessing prior to simulation. Volumetric (voxel) data was from the transducer map (starting pressure binary map) and medium data output by k-Wave.
(PNG)

**S17 Fig. Skull masks (examples).** Examples of skull masks used in acoustic simulation over-layed onto their respective subject-specific structural MRIs. Slices are the same as those shown in Fig 7. Masks shown are those used during simulation (i.e. after full skull processing including upscaling and smoothing via morphological image processing). Slices shown were made at the maximum pressure value (as visible in Fig 7). Note that these are not standard slices (i.e. coronal, sagittal, horizontal), since the volume was reoriented as part of pre-simulation processing.
(PDF)

**S18 Fig. tUS trajectory choice and tUS exposure protocol (location).** An illustration of the trajectory search protocol and tUS exposure protocol shared by both Experiment 1 and Experiment 2. (a) The search protocol used to find which three trajectories to use for tUS within the search grid (S12 Fig). For a search session (illustrative data), the mean TMS MEP trace generated at each trajectory is shown next to its corresponding grid location. The targets that elicited the largest, second-largest, and third-largest average MEPs were used as placement points for the NIBS devices. These three positions are referred to as "TMS target" (blue box), "2nd-best" (yellow box), and "3rd-best" (green box) targets respectively. (b) Illustration of the tUS burst durations per tUS trajectory. For full tUS protocol, see Fig 1.
(PDF)

**S19 Fig. Participant NIBS visit timelines.** Sequence of events for acquisition of resting TMS-evoked MEPs and tUS exposure. Not included in this diagram is the acquisition of TMS-evoked cSPs (*TMS cSPs*), which were acquired before all other experimental sections in all cases. (a) Timeline for the two volunteers who participated in only Experiment 1. The four blocks of 20 tUS trials each during tonic contraction paradigm: 300 ms at TMS target, 300 ms at 2$^{nd}$-best, 300 ms at 3$^{rd}$-best, 500 ms at TMS target. See *Experiment 1*, *cSPs* for full protocol. (b) Timeline for the eight volunteers who participated in both Experiment 1 and Experiment 2. 20 TMS-evoked MEPs were acquired both before and after the tUS exposure protocol. See *Experiment 2*, *Cortical excitability* for full protocol.
(PDF)

**S1 Table. Simulated pressure values for each trajectory used with ultrasound.** Data for both experiments included. Included are full width half maximum (FWHM) values [mm] of width

of the ellipsoid focus of the focused ultrasound beam. The maximum pressures for three key locations for the simulation are also shown: the maximum pressure anywhere, at $M1_{hand}$, and at the target coordinate used to aim the trajectory. In some cases, the trajectory coordinate and the $M1_{hand}$ coordinate are the same ('Target' column).
(PDF)

**S2 Table. K-Wave parameters.** Acoustic properties for media represented in k-Wave.
(PDF)

**S1 Text. Full screening questionnaire used for recruitment (neurological health).** A 'Yes' to any question prevented inclusion in the study.
(PDF)

**S1 File. Exposure formula.** An estimate of a single participant's cumulative $M1_{hand}$ exposure was made by multiplying the individual peak pressure at the $M1_{hand}$ voxel for all tUS trajectories by the time the tUS device was on for that location.
(PDF)

## Author Contributions

**Conceptualization:** Ian S. Heimbuch, Allan D. Wu, Andrew C. Charles, Marco Iacoboni.

**Data curation:** Ian S. Heimbuch, Tiffany K. Fan.

**Formal analysis:** Ian S. Heimbuch.

**Funding acquisition:** Andrew C. Charles.

**Investigation:** Ian S. Heimbuch, Tiffany K. Fan.

**Methodology:** Ian S. Heimbuch, Tiffany K. Fan, Guido C. Faas.

**Project administration:** Ian S. Heimbuch, Marco Iacoboni.

**Resources:** Marco Iacoboni.

**Software:** Ian S. Heimbuch, Allan D. Wu, Guido C. Faas.

**Supervision:** Allan D. Wu, Andrew C. Charles, Marco Iacoboni.

**Visualization:** Ian S. Heimbuch.

**Writing – original draft:** Ian S. Heimbuch, Tiffany K. Fan, Marco Iacoboni.

**Writing – review & editing:** Ian S. Heimbuch, Tiffany K. Fan, Allan D. Wu, Guido C. Faas, Andrew C. Charles, Marco Iacoboni.

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
