## [Decision Letter · Decision Letter 0]

15 Nov 2021

PONE-D-21-31823Ultrasound stimulation of the motor cortex during tonic muscle contractionPLOS ONE

Dear Dr. Heimbuch,

Thank you for submitting your manuscript to PLOS ONE. After careful consideration, we feel that it has merit but does not fully meet PLOS ONE’s publication criteria as it currently stands. Therefore, we invite you to submit a revised version of the manuscript that addresses the points raised during the review process.

Both reviewers provided detailed comments that need to be addressed. There should be more throughout discussion of relevant published papers (two were pointed out by the reviewers). The rationale for Experiment 2 as to why the chosen parameters are expected to produce offline effects should be more clearly stated. There is significant limitation with the low number of subjects which should be acknowledged or increase the number of subjects.

We look forward to receiving your revised manuscript.

Kind regards,

Robert Chen

Academic Editor

PLOS ONE

Journal Requirements:

Reviewers' comments:

Reviewer's Responses to Questions

**Comments to the Author**

1. Is the manuscript technically sound, and do the data support the conclusions?

Reviewer #1: Yes

Reviewer #2: Partly

2. Has the statistical analysis been performed appropriately and rigorously? 

Reviewer #1: Yes

Reviewer #2: Yes

3. Have the authors made all data underlying the findings in their manuscript fully available?

Reviewer #1: Yes

Reviewer #2: No

4. Is the manuscript presented in an intelligible fashion and written in standard English?

Reviewer #1: Yes

Reviewer #2: No

5. Review Comments to the Author

Reviewer #1: Please see attached document for my review.

Reviewer #2: Heimbuch et al., examine the effect of TUS of M1 on corticospinal excitability and inhibition as measured by EMG. In Experiment 1 TUS was applied during (i.e., “online”) voluntary muscle contraction and EMG was assessed with a focus on the cortical silent period (CSP), compared to TMS induced CSPs. In Experiment 2, TMS motor evoked potentials (MEP) were measured before and after the same TUS protocol applied at rest to assess after-effects (i.e., “offline” changes) in corticospinal excitability. While the justification for Experiment 1 is clear, the justification for the protocol in Experiment 2 (with relatively long inter-trial intervals) is unclear, as there is no indication in the literature that such a protocol with TUS (nor TMS) would result in offline effects that outlast the stimulation. Nonetheless, Experiment 1 provides interesting null results about the (lack of) TUS effects on inhibitory interneurons at previously used parameters, which are relevant for the TUS community. However, the methods section requires several clarifications, and a number of statements seem to oversimplify conclusions from previous literature. I have the following specific comments.

1. Line 64 – we don’t know whether these intensities are unsafe. They just haven’t been used in humans yet.

2. Lines 61 to 69 – would be good to cite Fomenko et al. 2020 and Xia et al. 2021 which also looked at TUS effects on M1.

3. Lines 70 to 77 – it would be helpful to discuss the effects (or lack thereof) of TUS on SICI (Legon et al. 2018 and Fomenko et al. 2020).

4. Lines 81 to 82 – better to state explicitly here already that Experiment 2 is about the after-effects (offline effects) of TUS

5. In Experiment 1, was the order of TUS vs TMS blocks counterbalanced? Were there any sham/ control trials between TUS or TMS trials?

6. Lines 105-115 – It sounds like the motor hotspot search was entirely based on a couple for predefined targets and not really optimized iteratively to find the best coil position and orientation for good, stable, and well-formed MEPs. Is it possible that the optimal TME MEP hotspot was not determined and application of them more focal TUS at those locations was therefore ineffective? Please show the MEP waveform averages for all subjects in the supplement.

7. Lines 129 to 136 – was the TMS-induced MEP measured before and after an entire block of 20 TUS trials?

8. Lines 137 to 144 – What specific coupling medium was used (gel, gel pads, water bladder, etc.) and how was good coupling ensured in the presence of hair?

9. Line 139 – is the focus 3 cm long, or 3cm away from the transducer face? Please provide both values, in addition to the width of the focus.

10. Line 153 – what attenuation factor is used to estimate in the intracranial intensity? Was it the equation on line 254? Please also provide the Ispta.

11. Lines 169 to 182 – the descriptions of cSP onset and offset determination is difficult to understand. For instance, why would the ‘first rising EMG value’ be used as onset? Please try to use a figure to explain this. And also mention whether similar methods have been used in other papers.

12. Line 189 – why the mean of the two flanking valleys instead of the larger valley? Does this method have any advantages, and has it been previously used?

13. Line 210 – what is the criteria for determining ‘silence’? Is it different from the threshold used to determine cSP onset and offset?

14. Please specify the exact MR acquisition parameters for the anatomical MRIs that were used for neuronavigation and acoustic simulation. Please also provide examples of the skull masks overlayed on the subject specific MRIs.

15. Line 279 to 282 – what about the TUS trials?

16. Line 283 – can you please elaborate how ‘spike rate’ is measured? Is this the ‘rate of EMG peaks’ described earlier? Does this measure have any specific physiological significance?

17. Line 396 – it is likely that some early studies recorded responses to the acoustic rather than neuromodulatory effects of TUS. However, recent papers have confirmed that neuromodulation is also present (see for eg. Mohammadjavadi et al. 2019, Brain Stimulation). I recommend adding some nuance to this statement. On the topic of sensory confounds, did any participants in your study report auditory and/ or somatosensory perception accompanying the TUS trials? Have they been asked explicitly?

18. Lines 408 to 409 – again, I recommend adding some nuance to this statement. While TUS may preferentially stimulate inhibitory neurons at some parameter combinations, there is also evidence that it can stimulate excitatory neurons (see Yu et al. 2021, Nature Communications).

19. Fig. S3 – it seems like this figure shows fewer peaks in the TMS compared to TUS trials. I presume the amplitude of the peaks where higher in TMS?

20. Fig. 5 – Why are MEP AUCs shown on the y-axis as if they only had a very limited number of discrete values? Please alternatively also show (and used for statistics) the more well-established MEP peak-to-peak amplitudes.

21. Fig. 7. – from the caption it does not become clear what the difference between (a) and (b) is supposed to be. In (a), both TUS beams seem to have “hit” the central sulcus in the hand knob, if anything more post-centrally, whereas in (b) it looks like the focal spot is very anterior and rather in the premotor cortex. None of the presented foci seem to be particularly well targeting the anterior wall of the central sulcus where M1 corticospinal output neurons are located. Simulated acoustic pressure maps for all targets and should be presented for all subjects in the Supplement.

22. Fig. 7 – it looks like the skull image for the subj-01 is not complete. Maybe the FOV was cut-off? This has major implications for the quality of the simulations. Please discuss these limitations.

23. Some information seem to be missing in the data availability statement: “All EMG files are available from the XXX database (url: XXX). Code to run acoustic simulations is available via the MATLAB package TUSX (www.tusx.org).”

6. PLOS authors have the option to publish the peer review history of their article (what does this mean?). If published, this will include your full peer review and any attached files.

Reviewer #1: No

Reviewer #2: No

---

## [Author Response · Author response to Decision Letter 0]

17 Jan 2022

(Full response attached as file)

We would like to thank the reviewers and the editor for their helpful feedback on the manuscript. Your

feedback was thorough, including both focused points to improve understanding of specific sections and

broader points to touch on questions concerning the field. In particular, we would like to thank you all for

bringing Fomenko, Chen et al., 2020 further into our attention.

We feel that your feedback has allowed us to improve clarity throughout the manuscript. Additionally, it has

helped us more fully cover details that tUS researchers may wish to know about our work, as well as

strengthening the manuscript’s scientific discussion. For that, we are truly appreciative.

---

## [Decision Letter · Decision Letter 1]

1 Mar 2022

PONE-D-21-31823R1Ultrasound stimulation of the motor cortex during tonic muscle contractionPLOS ONE

Dear Dr. Heimbuch,

Thank you for submitting your manuscript to PLOS ONE. After careful consideration, we feel that it has merit but does not fully meet PLOS ONE’s publication criteria as it currently stands. Therefore, we invite you to submit a revised version of the manuscript that addresses the points raised during the review process.

Both reviewers felt that the paper has improved but there are still points that needs to be addressed. The control experiment is reasonable. The experiment should be performed or at a minimum the issues discussed.

We look forward to receiving your revised manuscript.

Kind regards,

Robert Chen

Academic Editor

PLOS ONE

Journal Requirements:

Reviewers' comments:

Reviewer's Responses to Questions

**Comments to the Author**

1. If the authors have adequately addressed your comments raised in a previous round of review and you feel that this manuscript is now acceptable for publication, you may indicate that here to bypass the “Comments to the Author” section, enter your conflict of interest statement in the “Confidential to Editor” section, and submit your "Accept" recommendation.

Reviewer #1: All comments have been addressed

Reviewer #2: (No Response)

2. Is the manuscript technically sound, and do the data support the conclusions?

Reviewer #1: Yes

Reviewer #2: Yes

3. Has the statistical analysis been performed appropriately and rigorously? 

Reviewer #1: Yes

Reviewer #2: Yes

4. Have the authors made all data underlying the findings in their manuscript fully available?

Reviewer #1: Yes

Reviewer #2: Yes

5. Is the manuscript presented in an intelligible fashion and written in standard English?

Reviewer #1: Yes

Reviewer #2: Yes

6. Review Comments to the Author

Reviewer #1: Review was uploaded as an attachment.

Reviewer #2: Tha authors have adequately addressed most of my concerns. I have only some minor comments remaining:

1. The heading exposure vs. excitability is confusing. A suggestion is to say ‘association between exposure and excitability’ instead.

2. Can you please provide some supporting evidence for this statement –

‘In a scenario of on ongoing voluntary motor activity, a subset of interneurons in M1 will have already been recruited, so perhaps the channels or cells that tUS would have affected are already being actively modulated by the time tUS energies into the tissue (e.g. channels have already been opened).’

For instance, is there evidence from TMS studies suggesting that inhibitory interneurons are already active during a voluntary muscle contraction?

3. Please include the information about counter-balancing and sham in the manuscript.

7. PLOS authors have the option to publish the peer review history of their article (what does this mean?). If published, this will include your full peer review and any attached files.

Reviewer #1: No

Reviewer #2: No

---

## [Author Response · Author response to Decision Letter 1]

28 Mar 2022

We would like to thank the reviewers and the editor for their helpful feedback on the revised manuscript.

Responses are uploaded in ResponsetoReviewers_Revision2.docx

---

## [Editor Report · Decision Letter 2]

6 Apr 2022

Ultrasound stimulation of the motor cortex during tonic muscle contraction

PONE-D-21-31823R2

Dear Dr. Heimbuch,

We’re pleased to inform you that your manuscript has been judged scientifically suitable for publication and will be formally accepted for publication once it meets all outstanding technical requirements.

Kind regards,

Robert Chen

Academic Editor

PLOS ONE
---

## [Editor Report · Acceptance letter]

11 Apr 2022

PONE-D-21-31823R2 

Ultrasound stimulation of the motor cortex during tonic muscle contraction 

Dear Dr. Heimbuch:

I'm pleased to inform you that your manuscript has been deemed suitable for publication in PLOS ONE. Congratulations! Your manuscript is now with our production department. 

Kind regards, 

on behalf of

Dr. Robert Chen 

Academic Editor

PLOS ONE